# A Boosting Framework on Grounds of Online Learning

**Tofigh Naghibi, Beat Pfister**
Computer Engineering and Networks Laboratory
ETH Zurich, Switzerland
naghibi@tik.ee.ethz.ch, pfister@tik.ee.ethz.ch

## Abstract

By exploiting the duality between boosting and online learning, we present a boosting framework which proves to be extremely powerful thanks to employing the vast knowledge available in the online learning area. Using this framework, we develop various algorithms to address multiple practically and theoretically interesting questions including sparse boosting, smooth-distribution boosting, agnostic learning and, as a by-product, some generalization to double-projection online learning algorithms.

## 1 Introduction

A boosting algorithm can be seen as a meta-algorithm that maintains a distribution over the sample space. At each iteration a weak hypothesis is learned and the distribution is updated, accordingly. The output (strong hypothesis) is a convex combination of the weak hypotheses. Two dominant views to describe and design boosting algorithms are "weak to strong learner" (WTSL), which is the original viewpoint presented in [1, 2], and boosting by "coordinate-wise gradient descent in the functional space" (CWGD) appearing in later works [3, 4, 5]. A boosting algorithm adhering to the first view guarantees that it only requires a finite number of iterations (equivalently, finite number of weak hypotheses) to learn a $(1-\epsilon)$-accurate hypothesis. In contrast, an algorithm resulting from the CWGD viewpoint (usually called potential booster) may not necessarily be a boosting algorithm in the probability approximately correct (PAC) learning sense. However, while it is rather difficult to construct a boosting algorithm based on the first view, the algorithmic frameworks, e.g., AnyBoost [4], resulting from the second viewpoint have proven to be particularly prolific when it comes to developing new boosting algorithms. Under the CWGD view, the choice of the convex loss function to be minimized is (arguably) the cornerstone of designing a boosting algorithm. This, however, is a severe disadvantage in some applications.

In CWGD, the weights are not directly controllable (designable) and are only viewed as the values of the gradient of the loss function. In many applications, some characteristics of the desired distribution are known or given as problem requirements while, finding a loss function that generates such a distribution is likely to be difficult. For instance, what loss functions can generate sparse distributions?[1] What family of loss functions results in a smooth distribution?[2] We even can go further and imagine the scenarios in which a loss function needs to put more weights on a given subset of examples than others, either because that subset has more reliable labels or it is a problem requirement to have a more accurate hypothesis for that part of the sample space. Then, what

loss function can generate such a customized distribution? Moreover, does it result in a provable boosting algorithm? In general, how can we characterize the accuracy of the final hypothesis?

Although, to be fair, the so-called loss function hunting approach has given rise to useful boosting algorithms such as LogitBoost, FilterBoost, GiniBoost and MadaBoost [5, 8, 9, 10] which (to some extent) answer some of the above questions, it is an inflexible and relatively unsuccessful approach to addressing the boosting problems with distribution constraints.

Another approach to designing a boosting algorithm is to directly follow the WTSL viewpoint [11, 6, 12]. The immediate advantages of such an approach are, first, the resultant algorithms are provable boosting algorithms, i.e., they output a hypothesis of arbitrary accuracy. Second, the booster has direct control over the weights, making it more suitable for boosting problems subject to some distribution constraints. However, since the WTSL view does not offer any algorithmic framework (as opposed to the CWGD view), it is rather difficult to come up with a distribution update mechanism resulting in a provable boosting algorithm. There are, however, a few useful, and albeit fairly limited, algorithmic frameworks such as TotalBoost [13] that can be used to derive other provable boosting algorithms. The TotalBoost algorithm can maximize the margin by iteratively solving a convex problem with the totally corrective constraint. A more general family of boosting algorithms was later proposed by Shalev-Shwartz et. al. [14], where it was shown that weak learnability and linear separability are equivalent, a result following from von Neumann's minmax theorem. Using this theorem, they constructed a family of algorithms that maintain smooth distributions over the sample space, and consequently are noise tolerant. Their proposed algorithms find an $(1-\epsilon)$-accurate solution after performing at most $O(\log(N)/\epsilon^2)$ iterations, where $N$ is the number of training examples.

## 1.1 Our Results

We present a family of boosting algorithms that can be derived from well-known online learning algorithms, including projected gradient descent [15] and its generalization, mirror descent (both active and lazy updates, see [16]) and composite objective mirror descent (COMID) [17]. We prove the PAC learnability of the algorithms derived from this framework and we show that this framework in fact generates maximum margin algorithms. That is, given a desired accuracy level $\nu$, it outputs a hypothesis of margin $\gamma_{\min} - \nu$ with $\gamma_{\min}$ being the minimum edge that the weak classifier guarantees to return.

The duality between (linear) online learning and boosting is by no means new. This duality was first pointed out in [2] and was later elaborated and formalized by using the von Neumann's minmax theorem [18]. Following this line, we provide several proof techniques required to show the PAC learnability of the derived boosting algorithms. These techniques are fairly versatile and can be used to translate many other online learning methods into our boosting framework. To motivate our boosting framework, we derive two practically and theoretically interesting algorithms: (I) SparseBoost algorithm which by maintaining a sparse distribution over the sample space tries to reduce the space and the computation complexity. In fact this problem, i.e., applying batch boosting on the successive subsets of data when there is not sufficient memory to store an entire dataset, was first discussed by Breiman in [19], though no algorithm with theoretical guarantee was suggested. SparseBoost is the first provable batch booster that can (partially) address this problem. By analyzing this algorithm, we show that the tuning parameter of the regularization term $\ell_1$ at each round $t$ should not exceed $\frac{\gamma_t}{2}\eta_t$ to still have a boosting algorithm, where $\eta_t$ is the coefficient of the $t^{\text{th}}$ weak hypothesis and $\gamma_t$ is its edge. (II) A smooth boosting algorithm that requires only $O(\log 1/\epsilon)$ number of rounds to learn a $(1-\epsilon)$-accurate hypothesis. This algorithm can also be seen as an agnostic boosting algorithm[3] due to the fact that smooth distributions provide a theoretical guarantee for noise tolerance in various noisy learning settings, such as agnostic boosting [21, 22].

Furthermore, we provide an interesting theoretical result about MadaBoost [10]. We give a proof (to the best of our knowledge the only available unconditional proof) for the boosting property of (a variant of) MadaBoost and show that, unlike the common presumption, its convergence rate is of $O(1/\epsilon^2)$ rather than $O(1/\epsilon)$.

Finally, we show our proof technique can be employed to generalize some of the known online learning algorithms. Specifically, consider the Lazy update variant of the online Mirror Descent (LMD) algorithm (see for instance [16]). The standard proof to show that the LMD update scheme achieves vanishing regret bound is through showing its equivalence to the FTRL algorithm [16] in the case that they are both linearized, i.e., the cost function is linear. However, this indirect proof is fairly restrictive when it comes to generalizing the LMD-type algorithms. Here, we present a direct proof for it, which can be easily adopted to generalize the LMD-type algorithms.

## 2 Preliminaries

Let $\{(\mathbf{x}_i, a_i)\}, 1 \leq i \leq N$, be $N$ training samples, where $\mathbf{x}_i \in \mathcal{X}$ and $a_i \in \{-1, +1\}$. Assume $h \in \mathcal{H}$ is a real-valued function mapping $\mathcal{X}$ into $[-1, 1]$. Denote a distribution over the training data by $\mathbf{w} = [w_1, \ldots, w_N]^\top$ and define a loss vector $\mathbf{d} = [-a_1 h(\mathbf{x}_1), \ldots, -a_N h(\mathbf{x}_N)]^\top$. We define $\gamma = -\mathbf{w}^\top \mathbf{d}$ as the *edge* of the hypothesis $h$ under the distribution $w$ and it is assumed to be positive when $h$ is returned by a weak learner. In this paper we do not consider the branching program based boosters and adhere to the typical boosting protocol (described in Section 1).

Since a central notion throughout this paper is that of Bregman divergences, we briefly revisit some of their properties. A Bregman divergence is defined with respect to a convex function $\mathcal{R}$ as

$$B_\mathcal{R}(\mathbf{x}, \mathbf{y}) = \mathcal{R}(\mathbf{x}) - \mathcal{R}(\mathbf{y}) - \nabla \mathcal{R}(\mathbf{y})(\mathbf{x} - \mathbf{y})^\top \tag{1}$$

and can be interpreted as a distance measure between $\mathbf{x}$ and $\mathbf{y}$. Due to the convexity of $\mathcal{R}$, a Bregman divergence is always non-negative, i.e., $B_\mathcal{R}(\mathbf{x}, \mathbf{y}) \geq 0$. In this work we consider $\mathcal{R}$ to be a $\beta$-strongly convex function[4] with respect to a norm $||.||$. With this choice of $\mathcal{R}$, the Bregman divergence $B_\mathcal{R}(\mathbf{x}, \mathbf{y}) \geq \frac{\beta}{2}||\mathbf{x} - \mathbf{y}||^2$. As an example, if $\mathcal{R}(\mathbf{x}) = \frac{1}{2}\mathbf{x}^\top \mathbf{x}$ (which is 1-strongly convex with respect to $||.||_2$), then $B_\mathcal{R}(\mathbf{x}, \mathbf{y}) = \frac{1}{2}||\mathbf{x} - \mathbf{y}||_2^2$ is the Euclidean distance. Another example is the negative entropy function $\mathcal{R}(\mathbf{x}) = \sum_{i=1}^N x_i \log x_i$ (resulting in the KL-divergence) which is known to be 1-strongly convex over the probability simplex with respect to $\ell_1$ norm.

The Bregman projection is another fundamental concept of our framework.

**Definition 1 (Bregman Projection).** *The Bregman projection of a vector $\mathbf{y}$ onto a convex set $\mathcal{S}$ with respect to a Bregman divergence $B_\mathcal{R}$ is*

$$\Pi_\mathcal{S}(\mathbf{y}) = \underset{\mathbf{x} \in \mathcal{S}}{\arg\min}\, B_\mathcal{R}(\mathbf{x}, \mathbf{y}) \tag{2}$$

Moreover, the following generalized Pythagorean theorem holds for Bregman projections.

**Lemma 1 (Generalized Pythagorean)** [23, Lemma 11.3]. *Given a point $\mathbf{y} \in \mathbb{R}^N$, a convex set $\mathcal{S}$ and $\hat{\mathbf{y}} = \Pi_\mathcal{S}(\mathbf{y})$ as the Bregman projection of $\mathbf{y}$ onto $\mathcal{S}$, for all $\mathbf{x} \in \mathcal{S}$ we have*

$$\text{Exact:} \quad B_\mathcal{R}(\mathbf{x}, \mathbf{y}) \geq B_\mathcal{R}(\mathbf{x}, \hat{\mathbf{y}}) + B_\mathcal{R}(\hat{\mathbf{y}}, \mathbf{y}) \tag{3}$$

$$\text{Relaxed:} \quad B_\mathcal{R}(\mathbf{x}, \mathbf{y}) \geq B_\mathcal{R}(\mathbf{x}, \hat{\mathbf{y}}) \tag{4}$$

The relaxed version follows from the fact that $B_\mathcal{R}(\hat{\mathbf{y}}, \mathbf{y}) \geq 0$ and thus can be ignored.

**Lemma 2.** *For any vectors $\mathbf{x}, \mathbf{y}, \mathbf{z}$, we have*

$$(\mathbf{x} - \mathbf{y})^\top (\nabla \mathcal{R}(\mathbf{z}) - \nabla \mathcal{R}(\mathbf{y})) = B_\mathcal{R}(\mathbf{x}, \mathbf{y}) - B_\mathcal{R}(\mathbf{x}, \mathbf{z}) + B_\mathcal{R}(\mathbf{y}, \mathbf{z}) \tag{5}$$

The above lemma follows directly from the Bregman divergence definition in (1). Additionally, the following definitions from convex analysis are useful throughout the paper.

**Definition 2 (Norm & dual norm).** *Let $||.||_A$ be a norm. Then its dual norm is defined as*

$$||\mathbf{y}||_{A^*} = \sup\{\mathbf{y}^\top \mathbf{x}, ||\mathbf{x}||_A \leq 1\} \tag{6}$$

For instance, the dual norm of $||.||_2 = \ell_2$ is $||.||_{2^*} = \ell_2$ norm and the dual norm of $\ell_1$ is $\ell_\infty$ norm. Further,

**Lemma 3.** *For any vectors $\mathbf{x}, \mathbf{y}$ and any norm $||.||_A$, the following inequality holds:*

$$\mathbf{x}^\top \mathbf{y} \leq ||\mathbf{x}||_A ||\mathbf{y}||_{A^*} \leq \frac{1}{2}||\mathbf{x}||_A^2 + \frac{1}{2}||\mathbf{y}||_{A^*}^2 \tag{7}$$

Throughout this paper, we use the shorthands $||.||_A = ||.||$ and $||.||_{A^*} = ||.||_*$ for the norm and its dual, respectively.

Finally, before continuing, we establish our notations. Vectors are lower case bold letters and their entries are non-bold letters with subscripts, such as $x_i$ of $\mathbf{x}$, or non-bold letter with superscripts if the vector already has a subscript, such as $x_t^i$ of $\mathbf{x}_t$. Moreover, an N-dimensional probability simplex is denoted by $\mathcal{S} = \{\mathbf{w} | \sum_{i=1}^N w_i = 1, w_i \geq 0\}$. The proofs of the theorems and the lemmas can be found in the Supplement.

## 3 Boosting Framework

Let $\mathcal{R}(\mathbf{x})$ be a 1-strongly convex function with respect to a norm $||.||$ and denote its associated Bregman divergence $B_\mathcal{R}$. Moreover, let the dual norm of a loss vector $\mathbf{d}_t$ be upper bounded, i.e., $||\mathbf{d}_t||_* \leq L$. It is easy to verify that for $\mathbf{d}_t$ as defined in MABoost, $L = 1$ when $||.||_* = \ell_\infty$ and $L = N$ when $||.||_* = \ell_2$. The following Mirror Ascent Boosting (MABoost) algorithm is our boosting framework.

---

**Algorithm 1:** Mirror Ascent Boosting (MABoost)

---

**Input:** $\mathcal{R}(\mathbf{x})$ 1-strongly convex function, $\mathbf{w}_1 = [\frac{1}{N}, \dots, \frac{1}{N}]^\top$ and $\mathbf{z}_1 = [\frac{1}{N}, \dots, \frac{1}{N}]^\top$

**For** $t = 1, \dots, T$ **do**

    (a)   Train classifier with $\mathbf{w}_t$ and get $h_t$, let $\mathbf{d}_t = [-a_1 h_t(\mathbf{x}_1), \dots, -a_N h_t(\mathbf{x}_N)]$ and $\gamma_t = -\mathbf{w}_t^\top \mathbf{d}_t$.

    (b)   Set $\eta_t = \frac{\gamma_t}{L}$

    (c)   Update weights:    $\nabla \mathcal{R}(\mathbf{z}_{t+1}) = \nabla \mathcal{R}(\mathbf{z}_t) + \eta_t \mathbf{d}_t$    (lazy update)

                                    $\nabla \mathcal{R}(\mathbf{z}_{t+1}) = \nabla \mathcal{R}(\mathbf{w}_t) + \eta_t \mathbf{d}_t$    (active update)

    (d)   Project onto $\mathcal{S}$:    $\mathbf{w}_{t+1} = \underset{\mathbf{w} \in \mathcal{S}}{\operatorname{argmin}}\, B_\mathcal{R}(\mathbf{w}, \mathbf{z}_{t+1})$

**End**

**Output:** The final hypothesis $f(\mathbf{x}) = \text{sign}\left( \sum_{t=1}^T \eta_t h_t(\mathbf{x}) \right)$.

---

This algorithm is a variant of the mirror descent algorithm [16], modified to work as a boosting algorithm. The basic principle in this algorithm is quite clear. As in ADABoost, the weight of a wrongly (correctly) classified sample increases (decreases). The weight vector is then projected onto the probability simplex in order to keep the weight sum equal to 1. The distinction between the active and lazy update versions and the fact that the algorithm may behave quite differently under different update strategies should be emphasized. In the lazy update version, the norm of the auxiliary variable $\mathbf{z}_t$ is unbounded which makes the lazy update inappropriate in some situations. In the active update version, on the other hand, the algorithm always needs to access (compute) the previous projected weight $\mathbf{w}_t$ to update the weight at round $t$ and this may not be possible in some applications (such as boosting-by-filtering).

Due to the duality between online learning and boosting, it is not surprising that MABoost (both the active and lazy versions) is a boosting algorithm. The proof of its boosting property, however, reveals some interesting properties which enables us to generalize the MABoost framework. In the following, only the proof of the active update is given and the lazy update is left to Section 3.4.

**Theorem 1.** *Suppose that MABoost generates weak hypotheses $h_1, \dots, h_T$ whose edges are $\gamma_1, \dots, \gamma_T$. Then the error $\epsilon$ of the combined hypothesis $f$ on the training set is bounded and yields for the following $\mathcal{R}$ functions*:

$$\mathcal{R}(\mathbf{w}) = \frac{1}{2}||\mathbf{w}||_2^2 : \qquad\qquad \epsilon \leq \frac{1}{\sum_{t=1}^T \frac{1}{2}\gamma_t^2 + 1} \qquad\qquad (8)$$

$$\mathcal{R}(\mathbf{w}) = \sum_{i=1}^N w_i \log w_i : \qquad\qquad \epsilon \leq e^{-\sum_{t=1}^T \frac{1}{2}\gamma_t^2} \qquad\qquad (9)$$

In fact, the first bound (8) holds for any 1-strongly convex $\mathcal{R}$, though for some $\mathcal{R}$ (e.g., negative entropy) a much tighter bound as in (9) can be achieved.

*Proof*: Assume $\mathbf{w}^* = [w_1^*, \ldots, w_N^*]^\top$ is a distribution vector where $w_i^* = \frac{1}{N\epsilon}$ if $f(\mathbf{x}_i) \neq a_i$, and 0 otherwise. $\mathbf{w}^*$ can be seen as a uniform distribution over the wrongly classified samples by the ensemble hypothesis $f$. Using this vector and following the approach in [16], we derive the upper bound of $\sum_{t=1}^T \eta_t(\mathbf{w}^{*\top}\mathbf{d}_t - \mathbf{w}_t^\top \mathbf{d}_t)$ where $\mathbf{d}_t = [d_t^1, \ldots, d_t^N]$ is a loss vector as defined in Algorithm 1.

$$(\mathbf{w}^* - \mathbf{w}_t)^\top \eta_t \mathbf{d}_t = (\mathbf{w}^* - \mathbf{w}_t)^\top \big(\nabla\mathcal{R}(\mathbf{z}_{t+1}) - \nabla\mathcal{R}(\mathbf{w}_t)\big) \tag{10a}$$

$$= B_{\mathcal{R}}(\mathbf{w}^*, \mathbf{w}_t) - B_{\mathcal{R}}(\mathbf{w}^*, \mathbf{z}_{t+1}) + B_{\mathcal{R}}(\mathbf{w}_t, \mathbf{z}_{t+1}) \tag{10b}$$

$$\leq B_{\mathcal{R}}(\mathbf{w}^*, \mathbf{w}_t) - B_{\mathcal{R}}(\mathbf{w}^*, \mathbf{w}_{t+1}) + B_{\mathcal{R}}(\mathbf{w}_t, \mathbf{z}_{t+1}) \tag{10c}$$

where the first equation follows Lemma 2 and inequality (10c) results from the relaxed version of Lemma 1. Note that Lemma 1 can be applied here because $\mathbf{w}^* \in \mathcal{S}$.

Further, the $B_{\mathcal{R}}(\mathbf{w}_t, \mathbf{z}_{t+1})$ term is bounded. By applying Lemma 3

$$B_{\mathcal{R}}(\mathbf{w}_t, \mathbf{z}_{t+1}) + B_{\mathcal{R}}(\mathbf{z}_{t+1}, \mathbf{w}_t) = (\mathbf{z}_{t+1} - \mathbf{w}_t)^\top \eta_t \mathbf{d}_t \leq \frac{1}{2}||\mathbf{z}_{t+1} - \mathbf{w}_t||^2 + \frac{1}{2}\eta_t^2 ||\mathbf{d}_t||_*^2 \tag{11}$$

and since $B_{\mathcal{R}}(\mathbf{z}_{t+1}, \mathbf{w}_t) \geq \frac{1}{2}||\mathbf{z}_{t+1} - \mathbf{w}_t||^2$ due to the 1-strongly convexity of $\mathcal{R}$, we have

$$B_{\mathcal{R}}(\mathbf{w}_t, \mathbf{z}_{t+1}) \leq \frac{1}{2}\eta_t^2 ||\mathbf{d}_t||_*^2 \tag{12}$$

Now, replacing (12) into (10c) and summing it up from $t = 1$ to $T$, yields

$$\sum_{t=1}^T \mathbf{w}^{*\top}\eta_t \mathbf{d}_t - \mathbf{w}_t^\top \eta_t \mathbf{d}_t \leq \sum_{t=1}^T \frac{1}{2}\eta_t^2 ||\mathbf{d}_t||_*^2 + B_{\mathcal{R}}(\mathbf{w}^*, \mathbf{w}_1) - B_{\mathcal{R}}(\mathbf{w}^*, \mathbf{w}_{T+1}) \tag{13}$$

Moreover, it is evident from the algorithm description that for mistakenly classified samples

$$-a_i f(\mathbf{x}_i) = -a_i \text{sign}\left(\sum_{t=1}^T \eta_t h_t(\mathbf{x}_i)\right) = \text{sign}\left(\sum_{t=1}^T \eta_t d_t^i\right) \geq 0 \quad \forall \mathbf{x}_i \in \{\mathbf{x}|f(\mathbf{x}_i) \neq a_i\} \tag{14}$$

Following (14), the first term in (13) will be $\mathbf{w}^{*\top} \sum_{t=1}^T \eta_t \mathbf{d}_t \geq 0$ and thus, can be ignored. Moreover, by the definition of $\gamma$, the second term is $\sum_{t=1}^T -\mathbf{w}_t^\top \eta_t \mathbf{d}_t = \sum_{t=1}^T \eta_t \gamma_t$. Putting all these together, ignoring the last term in (13) and replacing $||\mathbf{d}_t||_*^2$ with its upper bound $L$, yields

$$-B_{\mathcal{R}}(\mathbf{w}^*, \mathbf{w}_1) \leq L\sum_{t=1}^T \frac{1}{2}\eta_t^2 - \sum_{t=1}^T \eta_t \gamma_t \tag{15}$$

Replacing the left side with $-B_{\mathcal{R}} = -||\mathbf{w}^* - \mathbf{w}_1||^2 = \frac{\epsilon-1}{N\epsilon}$ for the case of quadratic $\mathcal{R}$, and with $-B_{\mathcal{R}} = \log(\epsilon)$ when $\mathcal{R}$ is a negative entropy function, taking the derivative w.r.t $\eta_t$ and equating it to zero (which yields $\eta_t = \frac{\gamma_t}{L}$) we achieve the error bounds in (8) and (9). Note that in the case of $\mathcal{R}$ being the negative entropy function, Algorithm 1 degenerates into ADABoost with a different choice of $\eta_t$.

Before continuing our discussion, it is important to mention that the cornerstone concept of the proof is the choice of $\mathbf{w}^*$. For instance, a different choice of $\mathbf{w}^*$ results in the following max-margin theorem.

**Theorem 2.** *Setting $\eta_t = \frac{\gamma_t}{L\sqrt{t}}$, MABoost outputs a hypothesis of margin at least $\gamma_{min} - \nu$, where $\nu$ is a desired accuracy level and tends to zero in $O(\frac{\log T}{\sqrt{T}})$ rounds of boosting.*

**Observations:** Two observations follow immediately from the proof of Theorem 1. First, the requirement of using Lemma 1 is $\mathbf{w}^* \in \mathcal{S}$, so in the case of projecting onto a smaller convex set $\mathcal{S}_k \subseteq \mathcal{S}$, as long as $\mathbf{w}^* \in \mathcal{S}_k$ holds, the proof is intact. Second, only the relaxed version of Lemma 1 is required in the proof (to obtain inequality (10c)). Hence, if there is an approximate projection operator $\hat{\Pi}_{\mathcal{S}}$ that satisfies the inequality $B_{\mathcal{R}}(\mathbf{w}^*, \mathbf{z}_{t+1}) \geq B_{\mathcal{R}}\big(\mathbf{w}^*, \hat{\Pi}_{\mathcal{S}}(\mathbf{z}_{t+1})\big)$, it can be substituted

for the exact projection operator $\Pi_{\mathcal{S}}$ and the active update version of the algorithm still works. A practical approximate operator of this type can be obtained by using the double-projection strategy as in Lemma 4.

**Lemma 4.** *Consider the convex sets $\mathcal{K}$ and $\mathcal{S}$, where $\mathcal{S} \subseteq \mathcal{K}$. Then for any $\mathbf{x} \in \mathcal{S}$ and $\mathbf{y} \in \mathbb{R}^N$, $\hat{\Pi}_{\mathcal{S}}(\mathbf{y}) = \Pi_{\mathcal{S}}\left(\Pi_{\mathcal{K}}(\mathbf{y})\right)$ is an approximate projection operator that satisfies $B_{\mathcal{R}}(\mathbf{x}, \mathbf{y}) \geq B_{\mathcal{R}}\left(\mathbf{x}, \hat{\Pi}_{\mathcal{S}}(\mathbf{y})\right)$.*

These observations are employed to generalize Algorithm 1. However, we want to emphasis that the approximate Bregman projection is only valid for the active update version of MABoost.

### 3.1 Smooth Boosting

Let $k > 0$ be a smoothness parameter. A distribution $\mathbf{w}$ is smooth w.r.t a given distribution $\mathbf{D}$ if $w_i \leq kD_i$ for all $1 \leq i \leq N$. Here, we consider the smoothness w.r.t to the uniform distribution, i.e., $D_i = \frac{1}{N}$. Then, given a desired smoothness parameter $k$, we require a boosting algorithm that only constructs distributions $\mathbf{w}$ such that $w_i \leq \frac{k}{N}$, while guaranteeing to output a $(1 - \frac{1}{k})$-accurate hypothesis. To this end, we only need to replace the probability simplex $\mathcal{S}$ with $\mathcal{S}_k = \{\mathbf{w} | \sum_{i=1}^N w_i = 1, 0 \leq w_i \leq \frac{k}{N}\}$ in MABoost to obtain a smooth distribution boosting algorithm, called smooth-MABoost. That is, the update rule is: $\mathbf{w}_{t+1} = \underset{\mathbf{w} \in \mathcal{S}_k}{\operatorname{argmin}} B_{\mathcal{R}}(\mathbf{w}, \mathbf{z}_{t+1})$.

Note that the proof of Theorem 1 holds for smooth-MABoost, as well. As long as $\epsilon \geq \frac{1}{k}$, the error distribution $\mathbf{w}^*$ ($w_i^* = \frac{1}{N\epsilon}$ if $f(\mathbf{x}_i) \neq a_i$, and 0 otherwise) is in $\mathcal{S}_k$ because $\frac{1}{N\epsilon} \leq \frac{k}{N}$. Thus, based on the first observation, the error bounds achieved in Theorem 1 hold for $\epsilon \geq \frac{1}{k}$. In particular, $\epsilon = \frac{1}{k}$ is reached after a finite number of iterations. This projection problem has already appeared in the literature. An entropic projection algorithm ($\mathcal{R}$ is negative entropy), for instance, was proposed in [14]. Using negative entropy and their suggested projection algorithm results in a fast smooth boosting algorithm with the following convergence rate.

**Theorem 3.** *Given $\mathcal{R}(\mathbf{w}) = \sum_{i=1}^N w_i \log w_i$ and a desired $\epsilon$, smooth-MABoost finds a $(1 - \epsilon)$-accurate hypothesis in $O(\log(\frac{1}{\epsilon})/\gamma^2)$ of iterations.*

### 3.2 Combining Datasets

Let's assume we have two sets of data. A primary dataset $\mathcal{A}$ and a secondary dataset $\mathcal{B}$. The goal is to train a classifier that achieves $(1 - \epsilon)$ accuracy on $\mathcal{A}$ while limiting the error on dataset $\mathcal{B}$ to $\epsilon_{\mathcal{B}} \leq \frac{1}{k}$. This scenario has many potential applications including transfer learning [24], weighted combination of datasets based on their noise level and emphasizing on a particular region of a sample space as a problem requirement (e.g., a medical diagnostic test that should not make a wrong diagnosis when the sample is a pregnant woman). To address this problem, we only need to replace $\mathcal{S}$ in MABoost with $\mathcal{S}_c = \{\mathbf{w} | \sum_{i=1}^N w_i = 1, 0 \leq w_i \ \forall i \in \mathcal{A} \ \wedge \ 0 \leq w_i \leq \frac{k}{N} \ \forall i \in \mathcal{B}\}$ where $i \in \mathcal{A}$ shorthands the indices of samples in $\mathcal{A}$. By generating smooth distributions on $\mathcal{B}$, this algorithm limits the weight of the secondary dataset, which intuitively results in limiting its effect on the final hypothesis. The proof of its boosting property is quite similar to Theorem 1 and can be found in the Supplement.

### 3.3 Sparse Boosting

Let $\mathcal{R}(\mathbf{w}) = \frac{1}{2}||\mathbf{w}||_2^2$. Since in this case the projection onto the simplex is in fact an $\ell_1$-constrained optimization problem, it is plausible that some of the weights are zero (sparse distribution), which is already a useful observation. To promote the sparsity of the weight vector, we want to directly regularize the projection with the $\ell_1$ norm, i.e., adding $||\mathbf{w}||_1$ to the objective function in the projection step. It is, however, not possible in MABoost, since $||\mathbf{w}||_1$ is trivially constant on the simplex. Therefore, following the second observation, we split the projection step into two consecutive projections. The first projection is onto $\mathcal{K}$, an $N$-dimensional unit hypercube $\mathcal{K} = \{\mathbf{y} | 0 \leq y_i \leq 1\}$. This projection is regularized with the $\ell_1$ norm and the solution is then projected onto a simplex. Note

that the second projection can only make the solution sparser (look at the projection onto simplex algorithm in [25]).

---

**Algorithm 2:** SparseBoost

Let $\mathcal{K}$ be a hypercube and $\mathcal{S}$ a probability simplex; Set $\mathbf{w}_1 = [\frac{1}{N}, \ldots, \frac{1}{N}]^{\top}$;
At $t = 1, \ldots, T$, train $h_t$ with $\mathbf{w}_t$, set $\eta_t = \frac{\gamma_t}{N}$ and $0 \leq \alpha_t < \frac{\gamma_t}{2}$, and update

$$\mathbf{z}_{t+1} = \mathbf{w}_t + \eta_t \mathbf{d}_t$$

$$\mathbf{y}_{t+1} = \arg\min_{\mathbf{y} \in \mathcal{K}} ||\mathbf{y} - \mathbf{z}_{t+1}||^2 + \alpha_t \eta_t ||\mathbf{y}||_1$$

$$\mathbf{w}_{t+1} = \arg\min_{\mathbf{w} \in \mathcal{S}} ||\mathbf{w} - \mathbf{y}_{t+1}||^2$$

Output the final hypothesis $f(\mathbf{x}) = \text{sign}\left( \sum_{t=1}^{T} \eta_t h_t(\mathbf{x}) \right)$.

---

$\alpha_t$ is the regularization factor at round $t$. Since $\alpha_t \eta_t$ controls the sparsity of the solution, it is natural to investigate the maximum value that $\alpha_t$ can take, provided that the boosting property still holds. This bound is implicit in the following theorem.

**Theorem 4.** *Suppose that SparseBoost generates weak hypotheses $h_1, \ldots, h_T$ whose edges are $\gamma_1, \ldots, \gamma_T$. Then, as long as $\alpha_t \leq \frac{\gamma_t}{2}$, the error $\epsilon$ of the combined hypothesis $f$ on the training set is bounded as follows:*

$$\epsilon \leq \frac{1}{\sum_{t=1}^{T} \frac{1}{2} \gamma_t (\gamma_t - 2\alpha_t) + 1} \tag{16}$$

See the Supplement for the proof. It is noteworthy that SparseBoost can be seen as a variant of the COMID algorithm [17] with the difference that SparseBoost uses a double-projection or as called in Lemma 4, approximate projection strategy.

### 3.4 Lazy Update Boosting

In this section, we present the proof for the lazy update version of MABoost (LAMABoost) in Theorem 1. The proof technique is novel and can be used to generalize several known online learning algorithms such as OMDA in [26] and Meta algorithm in [27]. Moreover, we show that MadaBoost [10] can be presented in the LAMABoost setting. This gives a simple proof for MadaBoost without making the assumption that the edge sequence is monotonically decreasing (as in [10]).

*Proof*: Assume $\mathbf{w}^* = [w_1^*, \ldots, w_N^*]^{\top}$ is a distribution vector where $w_i^* = \frac{1}{N\epsilon}$ if $f(\mathbf{x}_i) \neq a_i$, and 0 otherwise. Then,

$$
\begin{aligned}
(\mathbf{w}^* - \mathbf{w}_t)^{\top} \eta_t \mathbf{d}_t &= (\mathbf{w}_{t+1} - \mathbf{w}_t)^{\top} \left( \nabla \mathcal{R}(\mathbf{z}_{t+1}) - \nabla \mathcal{R}(\mathbf{z}_t) \right) \\
&\quad + (\mathbf{z}_{t+1} - \mathbf{w}_{t+1})^{\top} \left( \nabla \mathcal{R}(\mathbf{z}_{t+1}) - \nabla \mathcal{R}(\mathbf{z}_t) \right) + (\mathbf{w}^* - \mathbf{z}_{t+1})^{\top} \left( \nabla \mathcal{R}(\mathbf{z}_{t+1}) - \nabla \mathcal{R}(\mathbf{z}_t) \right) \\
&\leq \frac{1}{2} ||\mathbf{w}_{t+1} - \mathbf{w}_t||^2 + \frac{1}{2} \eta_t^2 ||\mathbf{d}_t||_*^2 + B_{\mathcal{R}}(\mathbf{w}_{t+1}, \mathbf{z}_{t+1}) - B_{\mathcal{R}}(\mathbf{w}_{t+1}, \mathbf{z}_t) + B_{\mathcal{R}}(\mathbf{z}_{t+1}, \mathbf{z}_t) \\
&\quad - B_{\mathcal{R}}(\mathbf{w}^*, \mathbf{z}_{t+1}) + B_{\mathcal{R}}(\mathbf{w}^*, \mathbf{z}_t) - B_{\mathcal{R}}(\mathbf{z}_{t+1}, \mathbf{z}_t) \\
&\leq \frac{1}{2} ||\mathbf{w}_{t+1} - \mathbf{w}_t||^2 + \frac{1}{2} \eta_t^2 ||\mathbf{d}_t||_*^2 - B_{\mathcal{R}}(\mathbf{w}_{t+1}, \mathbf{w}_t) \\
&\quad + B_{\mathcal{R}}(\mathbf{w}_{t+1}, \mathbf{z}_{t+1}) - B_{\mathcal{R}}(\mathbf{w}_t, \mathbf{z}_t) - B_{\mathcal{R}}(\mathbf{w}^*, \mathbf{z}_{t+1}) + B_{\mathcal{R}}(\mathbf{w}^*, \mathbf{z}_t) \tag{17}
\end{aligned}
$$

where the first inequality follows applying Lemma 3 to the first term and Lemma 2 to the rest of the terms and the second inequality is the result of applying the exact version of Lemma 1 to $B_{\mathcal{R}}(\mathbf{w}_{t+1}, \mathbf{z}_t)$. Moreover, since $B_{\mathcal{R}}(\mathbf{w}_{t+1}, \mathbf{w}_t) - \frac{1}{2} ||\mathbf{w}_{t+1} - \mathbf{w}_t||^2 \geq 0$, they can be ignored in (17). Summing up the inequality (17) from $t = 1$ to $T$, yields

$$-B_{\mathcal{R}}(\mathbf{w}^*, \mathbf{z}_1) \leq L \sum_{t=1}^{T} \frac{1}{2} \eta_t^2 - \sum_{t=1}^{T} \eta_t \gamma_t \tag{18}$$

where we used the facts that $\mathbf{w}^{*\top} \sum_{t=1}^{T} \eta_t \mathbf{d}_t \geq 0$ and $\sum_{t=1}^{T} -\mathbf{w}_t^{\top} \eta_t \mathbf{d}_t = \sum_{t=1}^{T} \eta_t \gamma_t$. The above inequality is exactly the same as (15), and replacing $-B_{\mathcal{R}}$ with $\frac{\epsilon - 1}{N\epsilon}$ or $\log(\epsilon)$ yields the same

error bounds in Theorem 1. Note that, since the exact version of Lemma 1 is required to obtain (17), this proof does not reveal whether LAMABoost can be generalized to employ the double-projection strategy. In some particular cases, however, we may show that a double-projection variant of LAMABoost is still a provable boosting algorithm.

In the following, we briefly show that MadaBoost can be seen as a double-projection LAMABoost.

---

**Algorithm 3:** Variant of MadaBoost

---

Let $\mathcal{R}(\mathbf{w})$ be the negative entropy and $\mathcal{K}$ a unit hypercube; Set $\mathbf{z}_1 = [1, \dots, 1]^\top$;

At $t = 1, \dots, T$, train $h_t$ with $\mathbf{w}_t$, set $f_t(\mathbf{x}) = \mathrm{sign}\left( \sum_{t'=1}^{t} \eta_{t'} h_{t'}(\mathbf{x}) \right)$ and calculate

$\epsilon_t = \frac{\sum_{i=1}^{N} \frac{1}{2}|f_t(\mathbf{x}_i) - a_i|}{N}$, set $\eta_t = \epsilon_t \gamma_t$ and update

$$\nabla\mathcal{R}(\mathbf{z}_{t+1}) = \nabla\mathcal{R}(\mathbf{z}_t) + \eta_t \mathbf{d}_t \qquad\qquad \to z_{t+1}^i = z_t^i e^{\eta_t d_t^i}$$

$$\mathbf{y}_{t+1} = \underset{\mathbf{y}\in\mathcal{K}}{\arg\min} \, B_\mathcal{R}(\mathbf{y}, \mathbf{z}_{t+1}) \qquad\qquad \to y_{t+1}^i = \min(1, z_{t+1}^i)$$

$$\mathbf{w}_{t+1} = \underset{\mathbf{w}\in\mathcal{S}}{\arg\min} \, B_\mathcal{R}(\mathbf{w}, \mathbf{y}_{t+1}) \qquad\qquad \to w_{t+1}^i = \frac{y_{t+1}^i}{||\mathbf{y}_{t+1}||_1}$$

Output the final hypothesis $f(\mathbf{x}) = \mathrm{sign}\left( \sum_{t=1}^{T} \eta_t h_t(\mathbf{x}) \right)$.

---

Algorithm 3 is essentially MadaBoost, only with a different choice of $\eta_t$. It is well-known that the entropy projection onto the probability simplex results in the normalization and thus, the second projection of Algorithm 3. The entropy projection onto the unit hypercube, however, maybe less known and thus, its proof is given in the Supplement.

**Theorem 5.** *Algorithm 3 yields a* $(1-\epsilon)$-*accurate hypothesis after at most* $T = O(\frac{1}{\epsilon^2 \gamma^2})$.

This is an important result since it shows that MadaBoost seems, at least in theory, to be slower than what we hoped, namely $O(\frac{1}{\epsilon \gamma^2})$.

## 4   Conclusion and Discussion

In this work, we provided a boosting framework that can produce provable boosting algorithms. This framework is mainly suitable for designing boosting algorithms with distribution constraints. A sparse boosting algorithm that samples only a fraction of examples at each round was derived from this framework. However, since our proposed algorithm cannot control the exact number of zeros in the weight vector, a natural extension to this algorithm is to develop a boosting algorithm that receives the sparsity level as an input. However, this immediately raises the question: what is the maximum number of examples that can be removed at each round from the dataset, while still achieving a $(1-\epsilon)$-accurate hypothesis?

The boosting framework derived in this work is essentially the dual of the online mirror descent algorithm. This framework can be generalized in different ways. Here, we showed that replacing the Bregman projection step with the double-projection strategy, or as we call it approximate Bregman projection, still results in a boosting algorithm in the active version of MABoost, though this may not hold for the lazy version. In some special cases (MadaBoost for instance), however, it can be shown that this double-projection strategy works for the lazy version as well. Our conjecture is that under some conditions on the first convex set, the lazy version can also be generalized to work with the approximate projection operator. Finally, we provided a new error bound for the MadaBoost algorithm that does not depend on any assumption. Unlike the common conjecture, the convergence rate of MadaBoost (at least with our choice of $\eta$) is of $O(1/\epsilon^2)$.

### Acknowledgments

This work was partially supported by SNSF. We would like to thank Professor Rocco Servedio for an inspiring email conversation and our colleague Hui Liang for his helpful comments.

## Footnotes

[1] In the boosting terminology, sparsity usually refers to the greedy hypothesis-selection strategy of boosting methods in the functional space. However, sparsity in this paper refers to the sparsity of the distribution (weights) over the sample space.

[2] A smooth distribution is a distribution that does not put too much weight on any single sample or in other words, a distribution emulated by the booster does not dramatically diverge from the target distribution [6, 7].

[3]Unlike the PAC model, the agnostic learning model allows an arbitrary target function (labeling function) that may not belong to the class studied, and hence, can be viewed as a noise tolerant learning model [20].

[4]That is, its second derivative (Hessian in higher dimensions) is bounded away from zero by at least $\beta$.

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
