[Supplementary Material]

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

## Supplement

Before proceeding with the proofs, some definitions and facts need to be reminded.

### Definition: Margin

*Given a final hypothesis $f(\mathbf{x}) = \sum_{t=1}^{T} \eta_t h_t(\mathbf{x})$, the margin of a sample $(\mathbf{x}_j, a_j)$ is defined as $m(\mathbf{x}_j) = a_j f(\mathbf{x}_j)/\sum_{t=1}^{T} \eta_t$. Moreover, the margin of a set of examples denoted by $m_{\mathcal{D}}$ is the minimum of margins over the examples, i.e., $m_{\mathcal{D}} = \min_{\mathbf{x}} m(\mathbf{x}_j)$.*

### Fact: Duality between max-margin and min-edge

*The minimum edge $\gamma_{min}$ that can be achieved over all possible distributions of the training set is equal to the maximum margin ($m^* = \max_{\eta} m_{\mathcal{D}}$) of any linear combination of hypotheses from the hypotheses space.*

This fact is discussed in details in [28] and [29]. It is the direct result of von Neumann's minmax theorem and simply means that the maximum achievable margin is $\gamma_{min}$.

### Proof of Theorem 2

The proof for the maximum margin property of MABoost, is almost the same as the proof of Theorem 1.

Let's assume the $i^{\text{th}}$ sample has the worst margin, i.e., $m_{\mathcal{D}} = m(\mathbf{x}_i)$. Let all entries of the error vector $\mathbf{w}^*$ to be zero except its $i^{\text{th}}$ entry which is set to be 1. Following the same approach as in Theorem 1, (see equation (13)), we get

$$\sum_{t=1}^{T} \mathbf{w}^{*\top} \eta_t \mathbf{d}_t - \mathbf{w}_t^{\top} \eta_t \mathbf{d}_t \leq \sum_{t=1}^{T} \frac{1}{2} \eta_t^2 ||\mathbf{d}_t||_*^2 + B_{\mathcal{R}}(\mathbf{w}^*, \mathbf{w}_1) - B_{\mathcal{R}}(\mathbf{w}^*, \mathbf{w}_{T+1}) \qquad (19)$$

With our choice of $\mathbf{w}^*$ it is easy to verify that the first term on the left side of the inequality is $m_{\mathcal{D}} \sum_{t=1}^{T} \eta_t = -\sum_{t=1}^{T} \mathbf{w}^{*\top} \eta_t \mathbf{d}_t$. By setting $C = B_{\mathcal{R}}(\mathbf{w}^*, \mathbf{w}_1)$, ignoring the last term $B_{\mathcal{R}}(\mathbf{w}^*, \mathbf{w}_{T+1})$, replacing $||\mathbf{d}_t||_*^2$ with its upper bound $L$ and using the identity $\sum_{t=1}^{T} \mathbf{w}_t^{\top} \eta_t \mathbf{d}_t = -\sum_{t=1}^{T} \eta_t \gamma_t$ the above inequality is simplified to

$$-m_{\mathcal{D}} \sum_{t=1}^{T} \eta_t \leq L \sum_{t=1}^{T} \frac{1}{2} \eta_t^2 - \sum_{t=1}^{T} \eta_t \gamma_t + C \qquad (20)$$

Replacing $\eta_t$ with the value suggested in Theorem 2, i.e., $\eta_t = \dfrac{\gamma_t}{L\sqrt{t}}$ and dividing both sides by $\sum_{t=1}^{T} \eta_t$, gives

$$\frac{\sum_{t=1}^{T} (\frac{1}{\sqrt{t}} - \frac{1}{t})\gamma_t^2}{\sum_{t=1}^{T} \frac{1}{\sqrt{t}}\gamma_t} - \frac{LC}{\sum_{t=1}^{T} \frac{1}{\sqrt{t}}\gamma_t} \leq m_{\mathcal{D}} \qquad (21)$$

The first term is minimized when $\gamma_t = \gamma_{min}$. Similarly to the first term, the second term is maximized when $\gamma_t$ is replaced by its minimum value. This gives the following lower bound for $m_{\mathcal{D}}$:

$$\gamma_{min} \frac{\sum_{t=1}^{T} \frac{1}{\sqrt{t}} - \frac{1}{t}}{\sum_{t=1}^{T} \frac{1}{\sqrt{t}}} - \frac{LC}{\gamma_{min} \sum_{t=1}^{T} \frac{1}{\sqrt{t}}} \leq m_{\mathcal{D}} \qquad (22)$$

Considering the facts that $\int_1^{T+1} \frac{dx}{\sqrt{x}} \leq \sum_{t=1}^{T} \frac{1}{\sqrt{t}}$ and $1 + \int_1^T \frac{dx}{x} \geq \sum_{t=1}^{T} \frac{1}{t}$, we get

$$\gamma_{min} - \frac{1 + \log T}{2\sqrt{T+1} - 2}\gamma_{min} - \frac{LC}{\gamma_{min}(\sqrt{T+1} - 1)} \leq m_{\mathcal{D}} \qquad (23)$$

Now by taking $\nu = \frac{1+\log T}{2\sqrt{T+1}-2}\gamma_{min} + \frac{LC}{\gamma_{min}(\sqrt{T+1}-1)}$, we have $\gamma_{min} - \nu \leq \gamma_{min}$. It is clear from (23) that $\nu$ approaches zero as $T$ tends to infinity with a convergence rate proportional to $\frac{\log T}{\sqrt{T}}$. It is noteworthy that this convergence rate is slightly worse than of TotalBoost which is $O(\frac{1}{\sqrt{T}})$.

### Proof of Lemma 4

Remember that $\hat{\Pi}_{\mathcal{S}}(\mathbf{y}) = \Pi_{\mathcal{S}}\left(\Pi_{\mathcal{K}}(\mathbf{y})\right)$. Our goal is to show that $B_{\mathcal{R}}(\mathbf{x}, \mathbf{y}) \geq B_{\mathcal{R}}\left(\mathbf{x}, \hat{\Pi}_{\mathcal{S}}(\mathbf{y})\right)$.

To this end, we only need to repeatedly apply Lemma 1, as follows

$$B_{\mathcal{R}}(\mathbf{x}, \mathbf{y}) \geq B_{\mathcal{R}}\left(\mathbf{x}, \Pi_{\mathcal{K}}(\mathbf{y})\right) \qquad (24)$$

$$B_{\mathcal{R}}\left(\mathbf{x}, \Pi_{\mathcal{K}}(\mathbf{y})\right) \geq B_{\mathcal{R}}\left(\mathbf{x}, \hat{\Pi}_{\mathcal{S}}(\mathbf{y})\right) \qquad (25)$$

which completes the proof.

**Proof of combining datasets boosting algorithm**

We have to show that when the convex set is defined as

$$\mathcal{S}_c = \{\mathbf{w}| \sum_{i=1}^{N} w_i = 1, 0 \le w_i \ \ \forall i \in \mathcal{A} \ \wedge \ 0 \le w_i \le \frac{k}{N} \ \ \forall i \in \mathcal{B}\} \tag{26}$$

the error of the final hypothesis on $\mathcal{A}$, i.e., $\epsilon_{\mathcal{A}}$, converges to zero while the error on $\mathcal{B}$ is guaranteed to be $\epsilon_{\mathcal{B}} \le \frac{1}{k}$.

First, we show the convergence of $\epsilon_{\mathcal{A}}$ to zero. This is easily obtained by setting $\mathbf{w}^*$ to be an error vector with zero weights over the training samples from $\mathcal{B}$ and $\frac{1}{\epsilon_{\mathcal{A}} N_{\mathcal{A}}}$ weights over the training set $\mathcal{A}$. One can verify that $w^* \in \mathcal{S}_c$, thus the proof of Theorem 1 holds and subsequently, the error bounds in (8) stating that $\epsilon_{\mathcal{A}} \to 0$ as the number of iterations increases.

To show the second part of the theorem that is $\epsilon_{\mathcal{B}} \le \frac{1}{k}$, vector $\mathbf{w}^*$ is selected to be an error vector with zero weights over the training samples from $\mathcal{A}$ and $\frac{1}{\epsilon_{\mathcal{B}} N_{\mathcal{B}}}$ weights over the training set $\mathcal{B}$. Note that, as long as $\epsilon_{\mathcal{B}}$ is greater than $\frac{1}{k}$, this $w^* \in \mathcal{S}_c$. Thus, for all $\frac{1}{k} \le \epsilon_{\mathcal{B}}$ the proof of Theorem 1 holds and as the bounds in (8) show, the error decreases as the number of iterations increases. In particular in a finite number of rounds, the classification error on $\mathcal{B}$ reduces to $\frac{1}{k}$ which completes the proof.

**Proof of Theorem 4**

The proof is almost identical to the proof given in [17], with a slight change to take the double-projection strategy into account. Let $\mathbf{w}^*$ to be the same error vector as defined in Theorem 1. We start this proof by again bounding the

$$(\mathbf{w}^* - \mathbf{w}_t)^\top \eta_t \mathbf{d}_t = B_{\mathcal{R}}(\mathbf{w}^*, \mathbf{w}_t) - B_{\mathcal{R}}(\mathbf{w}^*, \mathbf{z}_{t+1}) + B_{\mathcal{R}}(\mathbf{w}_t, \mathbf{z}_{t+1}) \tag{27}$$

Remember that $\mathbf{y}_{t+1}$ is the solution of an optimization problem (Bregman projection). By writing the optimality condition at $\mathbf{y}_{t+1}$ we have

$$(\mathbf{w}^* - \mathbf{y}_{t+1})^\top (\nabla \mathcal{R}(\mathbf{y}_{t+1}) - \nabla \mathcal{R}(\mathbf{z}_{t+1})) + \alpha_t \eta_t \acute{r}(\mathbf{y}_{t+1})^\top (\mathbf{w}^* - \mathbf{y}_{t+1}) \ge 0 \tag{28}$$

where $\acute{r}(\mathbf{y})$ is a sub-gradient vector of the $\ell_1$ norm function $r(\mathbf{y}) = \sum_{i=1}^{N} y_i$. By applying the three point identity in Lemma 2, we get

$$B_{\mathcal{R}}(\mathbf{w}^*, \mathbf{z}_{t+1}) \ge B_{\mathcal{R}}(\mathbf{w}^*, \mathbf{y}_{t+1}) + B_{\mathcal{R}}(\mathbf{y}_{t+1}, \mathbf{z}_{t+1}) - \alpha_t \eta_t \acute{r}(\mathbf{y}_{t+1})^\top (\mathbf{w}^* - \mathbf{y}_{t+1}) \tag{29}$$

To bound the $\acute{r}(\mathbf{y}_{t+1})^\top (\mathbf{w}^* - \mathbf{y}_{t+1})$ in the above expression, we use the fact that $r(\mathbf{y})$ is a convex function and thus satisfies

$$\alpha_t \eta_t \acute{r}(\mathbf{y}_{t+1})^\top (\mathbf{w}^* - \mathbf{y}_{t+1}) \le \alpha_t \eta_t (r(\mathbf{w}^*) - r(\mathbf{y}_{t+1})) \tag{30}$$

Hence,

$$-B_{\mathcal{R}}(\mathbf{w}^*, \mathbf{z}_{t+1}) \le -B_{\mathcal{R}}(\mathbf{w}^*, \mathbf{y}_{t+1}) - B_{\mathcal{R}}(\mathbf{y}_{t+1}, \mathbf{z}_{t+1}) + \alpha_t \eta_t (r(\mathbf{w}^*) - r(\mathbf{y}_{t+1})) \tag{31}$$

Be applying Lemma 1 (generalized Pythagorean theorem) to $B_{\mathcal{R}}(\mathbf{w}_t, \mathbf{z}_{t+1})$ in (27), and replacing $B_{\mathcal{R}}(\mathbf{w}^*, \mathbf{z}_{t+1})$ with its upper bound from (31), we get the following bound:

$$\sum_{t=1}^{T} \mathbf{w}^{*\top} \eta_t \mathbf{d}_t \le \sum_{t=1}^{T} \mathbf{w}_t^\top \eta_t \mathbf{d}_t + \sum_{t=1}^{T} \frac{1}{2} \eta_t^2 ||\mathbf{d}_t||_*^2 + B_{\mathcal{R}}(\mathbf{w}^*, \mathbf{w}_1) + \sum_{t=1}^{T} \alpha_t \eta_t (1 - r(\mathbf{y}_{t+1})) \tag{32}$$

Now, replacing $r(\mathbf{y}_{t+1})$ with its lower bound, i.e, 0 and using the fact that $\sum_{t=1}^{T} \mathbf{w}^{*\top} \eta_t \mathbf{d}_t \ge 0$ (as shown in (14)) and $\sum_{t=1}^{T} \mathbf{w}_t^\top \eta_t \mathbf{d}_t = -\sum_{t=1}^{T} \eta_t \gamma_t$, yields

$$0 \le -\sum_{t=1}^{T} \eta_t \gamma_t + \sum_{t=1}^{T} \frac{1}{2} \eta_t^2 ||\mathbf{d}_t||_*^2 + B_{\mathcal{R}}(\mathbf{w}^*, \mathbf{w}_1) + \sum_{t=1}^{T} \alpha_t \eta_t \tag{33}$$

Since in SparseBoost $\mathcal{R}(\mathbf{w}) = \frac{1}{2} ||\mathbf{w}||_2^2$, the Bregman divergence $B_{\mathcal{R}}(\mathbf{w}^*, \mathbf{w}_1) = \frac{1-\epsilon}{N\epsilon}$ and $||\mathbf{d}_t||_*^2 \le N$. Replacing them in (33) and setting $\eta_t = \frac{\gamma_t}{N}$, gives the bound in Theorem 4.

**Proof of Entropy Projection onto Hypercube (Second Update Step in MadaBoost)**

**Lemma 5.** *Let $\mathcal{R}(\mathbf{w}) = \sum_{i=1}^{N} w_i \log w_i - w_i$. Then the Bregman projection of a positive vector $\mathbf{z} \in \mathbb{R}_+^N$ onto the unit hypercube $\mathcal{K} = [0, 1]^N$ is $y_i = \min(1, z_i), i = 1, \dots, N$.*

To show the correctness of the above lemma, i.e., that the solution of the Bregman projection

$$\mathbf{y} = \arg\min_{\mathbf{y} \in \mathcal{K}} B_{\mathcal{R}}(\mathbf{y}, \mathbf{z}) \tag{34}$$

is $y_i = \min(1, z_i)$, we only need to show that $\mathbf{y}$ satisfies the optimality condition

$$(\mathbf{v} - \mathbf{y})^\top \nabla B_{\mathcal{R}}(\mathbf{y}, \mathbf{z}) \geq 0 \quad \forall \mathbf{v} \in \mathcal{K} \tag{35}$$

Given $\mathcal{R}(\mathbf{w}) = \sum_{i=1}^{N} w_i \log w_i - w_i$, the gradient of $B_{\mathcal{R}}$ is

$$\nabla B_{\mathcal{R}}(\mathbf{y}, \mathbf{z}) = \sum_{i=1}^{T} \log \frac{y_i}{z_i} \tag{36}$$

Hence,

$$(\mathbf{v} - \mathbf{y})^\top \nabla B_{\mathcal{R}}(\mathbf{y}, \mathbf{z}) = \sum_{i \in \{i : z_i \geq 1\}} (v_i - y_i) \log \frac{y_i}{z_i} + \sum_{i \in \{i : z_i < 1\}} (v_i - y_i) \log \frac{y_i}{z_i} \tag{37}$$

For $z_i \geq 1$, $y_i$ is equal to 1. That is, $\log \frac{y_i}{z_i} = \log \frac{1}{z_i} < 0$. On the other hand, since $v_i \leq 1$, $(v_i - y_i) = (v_i - 1) \leq 0$. Thus, the first sum in (37) is always non-negative. The second sum is always zero since $\log \frac{y_i}{z_i} = \log 1 = 0$. That is, the optimality condition (37) is non-negative for all $\mathbf{v}$ which completes the proof.

**Proof of Theorem 5**

Its proof is essentially the same as the proof of the lazy version of MABoost with a few differences. Before proceeding further, some definitions and facts should be re-emphasized.

First of all, since $\mathcal{R}(\mathbf{w}) = \sum_{i=1}^{N} w_i \log w_i - w_i$ is $\frac{1}{N}$-strongly convex (see [30, p. 136]) with respect to $\ell_1$ norm (and not 1-strongly as in Theorem 1), the following inequality holds for the Bregman divergence:

$$B_{\mathcal{R}}(\mathbf{x}, \mathbf{y}) \geq \frac{1}{2N} ||\mathbf{x} - \mathbf{y}||_1^2 \tag{38}$$

Moreover, the following lemma which bounds $||\mathbf{y}_t||$ is essential for our proof.

**Lemma 6.** *For all $t$, $||\mathbf{y}_t||_1 \geq N\epsilon_t$ where $\epsilon_t$ is the error of the ensemble hypothesis $H_t(\mathbf{x}) = \sum_{l=1}^{t} \eta_l h_l(\mathbf{x})$ at round $t$.*

This lemma holds due to the fact that

$$y_t^i = \min(1, z_t^i) = \min(1, e^{\sum_{l=1}^{t} \eta_l d_l^i}) = \min(1, e^{-a_i H_t(\mathbf{x}_i)}) \tag{39}$$

where $H_t(\mathbf{x}) = \sum_{l=1}^{t} \eta_l h_l(\mathbf{x})$ is the output of the algorithm at round $t$. If $H_t(\mathbf{x}_i)$ makes a mistake on classifying $\mathbf{x}_i$, $-a_i H_t(\mathbf{x}_i)$ will be greater than zero and thus, $y_t^i = 1$. For the samples that are classified correctly, $-a_i H_t(\mathbf{x}_i) \leq 0$ and thus, $0 \leq y_t^i \leq 1$. That is, $N\epsilon_t$ = number of wrongly classified samples at round $t \leq \sum_{i=1}^{N} y_t^i = ||\mathbf{y}_t||_1$.

We are now ready to proceed with the proof of Theorem 5. Let $\mathbf{w}^* = [w_1^*, \cdots, w_N^*]^\top$ to be a vector where $w_i^* = 1$ if $f(\mathbf{x}_i) \neq a_i$, and 0 otherwise. Similar to the proof of the lazy update, we are going to bound the $\sum_{t=1}^{T} (\mathbf{w}^* - \mathbf{y}_t)^\top \eta_t \mathbf{d}_t$.

$$
\begin{aligned}
(\mathbf{w}^* - \mathbf{y}_t)^\top \eta_t \mathbf{d}_t &= (\mathbf{y}_{t+1} - \mathbf{y}_t)^\top \big(\nabla \mathcal{R}(\mathbf{z}_{t+1}) - \nabla \mathcal{R}(\mathbf{z}_t)\big) \\
&\quad + (\mathbf{z}_{t+1} - \mathbf{y}_{t+1})^\top \big(\nabla \mathcal{R}(\mathbf{z}_{t+1}) - \nabla \mathcal{R}(\mathbf{z}_t)\big) + (\mathbf{w}^* - \mathbf{z}_{t+1})^\top \big(\nabla \mathcal{R}(\mathbf{z}_{t+1}) - \nabla \mathcal{R}(\mathbf{z}_t)\big) \\
&\leq \frac{1}{2N} ||\mathbf{y}_{t+1} - \mathbf{y}_t||^2 + \frac{N}{2} \eta_t^2 ||\mathbf{d}_t||_*^2 + B_{\mathcal{R}}(\mathbf{y}_{t+1}, \mathbf{z}_{t+1}) - B_{\mathcal{R}}(\mathbf{y}_{t+1}, \mathbf{z}_t) + B_{\mathcal{R}}(\mathbf{z}_{t+1}, \mathbf{z}_t) \\
&\quad - B_{\mathcal{R}}(\mathbf{w}^*, \mathbf{z}_{t+1}) + B_{\mathcal{R}}(\mathbf{w}^*, \mathbf{z}_t) - B_{\mathcal{R}}(\mathbf{z}_{t+1}, \mathbf{z}_t) \\
&\leq \frac{1}{2N} ||\mathbf{y}_{t+1} - \mathbf{y}_t||^2 + \frac{N}{2} \eta_t^2 ||\mathbf{d}_t||_*^2 - B_{\mathcal{R}}(\mathbf{y}_{t+1}, \mathbf{y}_t) \\
&\quad + B_{\mathcal{R}}(\mathbf{y}_{t+1}, \mathbf{z}_{t+1}) - B_{\mathcal{R}}(\mathbf{y}_t, \mathbf{z}_t) - B_{\mathcal{R}}(\mathbf{w}^*, \mathbf{z}_{t+1}) + B_{\mathcal{R}}(\mathbf{w}^*, \mathbf{z}_t)
\end{aligned}
\tag{40}
$$

where the first inequality follows from applying Lemma 3 to the first term and Lemma 2 to the rest of the terms and the second inequality is the result of applying the exact version of Lemma 1 to $B_\mathcal{R}(\mathbf{y}_{t+1}, \mathbf{z}_t)$. Moreover, since according to inequality (38) $B_\mathcal{R}(\mathbf{y}_{t+1}, \mathbf{y}_t) - \frac{1}{2N}||\mathbf{y}_{t+1} - \mathbf{y}_t||^2 \geq 0$ and hence these terms can be ignored in (40). Summing up the inequality (40) from $t = 1$ to $T$, yields:

$$-B_\mathcal{R}(\mathbf{w}^*, \mathbf{z}_1) \leq \sum_{t=1}^{T} \frac{N}{2} \eta_t^2 - \sum_{t=1}^{T} \eta_t \gamma_t ||\mathbf{y}_t||_1 \qquad (41)$$

It is important to remark that $||\mathbf{y}_t||_1$ appearing in the last term is due to the fact that $\mathbf{w}_t = \frac{\mathbf{y}_t}{||\mathbf{y}_t||_1}$ and thus, $\mathbf{y}_t^\top \eta_t \mathbf{d}_t = \mathbf{w}_t^\top \eta_t \mathbf{d}_t ||\mathbf{y}_t||_1 = \eta_t \gamma_t ||\mathbf{y}_t||_1$.

Now, by replacing $\eta_t = \epsilon_t \gamma_t$ in the above equation and noting that $B_\mathcal{R}(\mathbf{w}^*, \mathbf{z}_1) = N - N\epsilon$, we get:

$$-N(1 - \epsilon) \leq \sum_{t=1}^{T} \frac{N}{2} \epsilon_t^2 \gamma_t^2 - \sum_{t=1}^{T} \epsilon_t \gamma_t^2 ||\mathbf{y}_t||_1 \qquad (42)$$

From Lemma 6, it is evident that $||\mathbf{y}_t||_1 \geq N\epsilon_t$. Moreover, since $\epsilon \leq \epsilon_t$, it can be replaced b $\epsilon$, as well (though very pessimistic). As usuall, $\gamma_t$ is also replaced with the min edge, denoted by $\gamma$. Applying these lower bounds in (42), yields

$$\epsilon^2 \leq \frac{2(1 - \epsilon)\gamma^2}{T} \leq \frac{\gamma^2}{T} \qquad (43)$$

which indicates that the proposed version of MadaBoost needs at most $O(\frac{1}{\epsilon^2 \gamma^2})$ iterations to converge.