[Reviews · NeurIPS 2014]

Submitted by Assigned_Reviewer_10

Summary:
This paper proposes boosting algorithms and analyze them from an online learning perspective. First, they propose a boosting algorithm based on the update of the online mirror descent(MABoost). Then they show a smooth version of MABoost, i.e., a variant of MABoost which creates only "smooth" distributions over examples. Further, they propose sparse or lazy versions of MABoost and show their convergence proofs. In particular, their result also implies an improved convergence bound for MadaBoost.

Comments:

The convergence proof of MadaBoost is new and improved w.r.t. the original version.

But, the analysis based on online learning (or using Bregman divergence) is not new. For example, for TotalBoost(ICML06) and SoftBoost(NIPS07), Warmuth et al already used such techniques to prove the convergence of the algorithms.

The author considers minimizing training error only, which is not an appropriate goal. Obviously, the goal should be to minimize the generalization error. Some of smooth boosting algorithms can be applied to the boosting-by-filtering framework in which the learner has an access to the example oracle returning a labeled example drawn randomly according to the underlying distribution and the goal is to directly minimize the generalization error w.r.t. D. See, e.g., the paper of MadaBoost, GiniBoost or FilterBoost for the details. But, the framework of the current paper does not seem to be applied to the filtering framework.

An alternative way to minimize the generalization error is to prove that the boosting algorithm optimizes the (soft) margin. TotalBoost, SoftBoost or ERLPBoost (Warmuth et al. ALT08) do this. The proposed methods, however, does not follow the approach as well.

In Theorem 1, the authors compare two versions of MABoost for two norm and relative entropy. Apparently, the bounds have exponential difference. But, the distribution created using relative entropy is much skewer in general and thus the edges of hypotheses become smaller in practice. The author should mention it, too.

Also, an experimental comparison is desirable.

Other minor comments
- There are some typos such as Appendix ??.

After viewing authors' comments:
I think the paper is OK if the authors add a discussion on generalization ability of the proposed methods in the revised version.
Summary: The analysis is interesting and it motivates new boosting algorithms. But some explanation or discussion on the generalization ability should be added.

Submitted by Assigned_Reviewer_30

The paper presents a new, quite general boosting algorithm based on mirror descent/ascent, together with the well-established connection between boosting and online learning. This result is then applied to various theoretical problems such as the derivation of a "smooth" boosting algorithm and an analysis of the MadaBoost algorithm.

As the paper acknowledges, the connection between boosting and online learning is not at all new. Nevertheless, the paper exploits this connection in novel ways that lead to interesting and significant new results. The analysis and algorithm are clean and general, and the applications very interesting. This seems like a significant advance in the development of general boosting-based methodology.

I think Theorem 1 is quite interesting. Even so, I am dubious about interpreting it to mean that this general algorithm is a boosting algorithm for the case given in Eq. (8). The bound does show that the training error epsilon goes to zero under the weak learning condition as T gets large. However, it appears to me that this bound is too weak to prove that the generalization error goes to zero, as required for strong PAC learning.

Related to this point, I would be curious to know what can be proved in general about the margins of the training examples for MABoost. If the margins can be shown to be large, then this would suffice to show that the algorithm is indeed a boosting algorithm.

The paper is entirely theoretical with no experimental results.

Other comments:

I very much liked the summary of the state of boosting research given in the introduction.

I think central concepts like smooth boosting should be described in the text, not in footnotes.

Page 2: I think another algorithmic framework for deriving boosting algorithms is given by the "drifting games" approach.

Line 105: I think the result in [13] only applies to a variant of LogitBoost.

Line 170: Should "x+i" be x_i?

Line 398: I'm not sure I understand the conclusion that Madaboost is slower than previously hoped. My understanding is that the new result (which is very nice) only proves an upper bound on this algorithm, not a lower bound. So isn't it still possible that a better upper bound could eventually be proved?

Line 415 (and elsewhere): "conjunction" -> "conjecture"

In general, the paper is badly in need of further proofreading, for instance, for numerous English mistakes and errors in the latex.
Summary: A significant advance in the development and theoretical understanding of boosting-based methodology.

Submitted by Assigned_Reviewer_36

This work describes an approach for obtaining boosting algorithms from the mirror-descent algorithm for (online) convex minimization. The algorithms that are derived have provable guarantees in the PAC framework (that is given a weak learner with advantage \gamma it produces an \epsilon-accurate hypothesis). The framework is useful when boosting needs to be performed with additional (convex) constraints on the distribution since those can be obtained by changing the Bregman projection step. In particular, smoothness and sparsity are analyzed. The paper also claims to contribute new analysis of mirror-descent with lazy updates.

This paper is a nice generalization of Kivinen-Warmuth 99 and Barak-Hardt-Kale 09 papers that only dealt with KL divergence to any Bregman divergence. The applications are nice but none of them seems both interesting and novel. Smoothness is certainly a useful property (e.g. for agnostic learning) but this case the approach to obtaining smoothness is identical to that in the BHK09 paper. In the case of sparsity the discussion of it is rather non-specific. No clear motivation and also no theoretical analysis of sparsity are given. The result only shows that it is possible to add an additional l_1 constraint into the boosting analysis. Without motivation and analysis it's impossible to tell for example how it would compare with subsampling.

At the technical level the results seem nice but rely primarily on the well-known connection of boosting and online learning/optimization and known analysis of mirror-descent. The paper mentions a number of "second-order" improvements to aspects of the analysis but I'm not familiar enough with those to judge the importance and novelty. Overall I think that the paper has interesting technical content which should be published. At the same time I expect more novelty (either conceptually or technically) for a solid recommendation to NIPS.

Some additional comments and typos:
053 - The term smooth boosting was first defined by Servedio, COLT 01 not Gavinsky (but smoothness was used even earlier by Jackson, FOCS 94)
095 - The bound on the number of steps lacks the \gamma_t term
098 - Smoothness is also shown to be useful for agnostic boosting by Gavinsky, ALT 02 and by Feldman, ICS 10 for agnostic and PAC learning models.
138 - It seems that later on you are using a different definition of entropy
156 - is -> are
169 - "norm of"
170 - typos in the definition of entropy
171 - what are x_i' here?
279 - 3 typos
415,419 conjunction->conjecture
441 - M.Long is not an author of this article (P.Long is the editor)
Summary: This paper presents a technique for obtaining boosting algorithms based on a generalization of some previous approaches. The results appear to be new and are useful to know but are technically fairly standard.
Author Feedback
Author rebuttal: We appreciate your encouraging and useful comments. The corrections you mentioned should be taken into account in the final version of the paper.

Answers to the 1st Reviewer:
As mentioned, MABOOST is presented in the boosting-by-sampling setting rather than in the boosting-by-filtering setting because: MABOOST is more general. Filtering setting only accepts a subset of smooth boosters while MABOOST as a framework can generate smooth and non smooth algorithms. In fact, some of the smooth boosters derived from MABOOST framework can be used in the filtering setting. For instance, using the sum of binary entropies as the Bregman divergence in the lazy version of MABOOST results in a logitboost-type smooth boosting algorithm. Following the techniques presented in Madaboost [Domingo2000] it can be straightforwardly converted to a boosting-by-filtering algorithm.

Another important point is (though not directly mentioned in the paper) MABOOST maximizes the minimum margin of the examples or in other words, it is a maximum margin framework. Thus, theoretically, it minimizes the upperbound of the generalization error. Admittedly, a line or two are needed to mention this property of MABOOST. It is quite easy to show it. In the proof of theorem 1, select w_* to be a vector with only one non-zero entry corresponding to the example with the worst margin (minimum margin over the examples). Then in inequality (13) the first term on the left is the margin and as can be seen with our choice of etha (etha_t=gamma_t/L), it converges to lambda_min (minimum edge). From [Rätsch & Warmuth,2005. Efficient...] or (first equation in TotalBoost) we know that lambda_min is the maximum margin which can be achieved. Thus, MABOOST is the maximum margin algorithm.

Eventually it is noteworthy that the analysis and the corresponding techniques used here such as choosing appropriate w_*, beta-strongly convex functions and double-Bregman projections are new in boosting context and has not been used before in Totalboost or other works. This is the first work that shows famous online learning algorithms are in fact boosting algorithms in a rigorous way.

Answers to the 2nd reviewer:

As explained above, MABOOST is a maximum margin framework and minimizes the upper bound of the generalization error. This important point should to be mentioned in the proof of theorem 1 after inequality 13. In fact, the first term on the left side of inequality (13) (normalized to the sum of etha_t) is the margin and by choosing a proper w_* it is obvious that this margin goes to its maximum value (min edge) as the number of rounds increases. Regarding line 398, as you noted, a better bound still may be found and the proposed bound is (probably) not tight. This has been mentioned in the conclusion.

Answers to the 3rd reviewer:
As mentioned, this paper can be considered as a generalization of [Kivinen-Warmuth, 99]. However, we want to point out that this generalization is not only in terms of using general Bregman divergences but also in terms of 1-establishing direct relation between famous online learning methods and boosting which can be used in practice and 2- using double-Bregman projection.

The motivations for sparse boosting may not have been stated clearly enough. Since batch boosting (boosting by sampling) must have access to the entire data set, it is rather difficult to apply it to very large datasets. This problem has been considered before in [L. Breiman, Pasting bites together....97]". A common way to address this issue is to use subsampling. First of all, the sparse boosting algorithm proposed here, can be used in combination with subsampling. That is, only a fraction of examples with non-zero weights are selected at each round. Second, we are not aware of any convergence analysis when subsampling is used in a boosting algorithm (although it could be our lack of knowledge) thus in case of having a large dataset with a requirement to satisfy PAC-learnability, we have to use sparse boosting instead of subsampling. Furthermore, as you mentioned, it is only imposing the l1 constraint and not a direct control over the number of zeros. It is an open problem to determine in advance how many of the weights can be set to zero while still having a boosting algorithm.

We also would like to use this opportunity to describe the main contributions of this work from our standpoint. We believe the first important contribution is that we've shown how online learning algorithms and proof techniques there can be used to construct useful provable boosting algorithms. Of course, the duality between online learning and boosting was known since the original Adaboost paper. Nevertheless, this connection had never been revisited with the currently available convex analysis techniques for online learning. This allows to import various useful algorithms from online learning to boosting area. For instance, [Langford - ‎2009 sparse online learning...] is a work in online learning that directly control the number of non-zero elements and with the techniques presented in our work, it may also be used in boosting. Using a self-concordant function as a loss instead of strongly-convex function as suggested in [Rakhlin COLT2013] is also another possibility of this type which may have its own benefits.
The second contribution of this work is to motivate to use multiple projections onto convex sets rather than directly projecting to a probability simplex. This may in fact result in some unexpected theoretical results such as finding an exponentially fast boosting-by-filtering method (or showing it is impossible) which is an open problem.